# Stereotactic Surgery for Treating Intractable Tourette Syndrome: A Single-Center Pilot Study

**DOI:** 10.3390/brainsci12070838

**Published:** 2022-06-28

**Authors:** Xin Wang, Liang Qu, Shunnan Ge, Nan Li, Jing Wang, Chun Qiu, Huijuan Kou, Jiaming Li, Jiangpeng Jing, Mingming Su, Zhaohui Zheng, Yang Li, Yan Qu, Xuelian Wang

**Affiliations:** 1Department of Neurosurgery, Tangdu Hospital, The Fourth Military Medical University, Xi’an 710038, China; wx2002@163.com (X.W.); quliang@fmmu.edu.cn (L.Q.); gesn8561@163.com (S.G.); linan314@fmmu.edu.cn (N.L.); wmwangj@163.com (J.W.); m18710930227@126.com (C.Q.); doc_leejm@163.com (J.L.); drjingjp@xjtufh.edu.cn (J.J.); summgs163-ok@163.com (M.S.); zzhzyq114@163.com (Z.Z.); sdbzliyang@126.com (Y.L.); yanqu0123@fmmu.edu.cn (Y.Q.); 2Department of Cardiology, The Second Affiliated Hospital, Xi’an Jiaotong University, Xi’an 710004, China; khjsarah@163.com

**Keywords:** stereotactic surgery, ablation, deep brain stimulation, Tourette syndrome

## Abstract

To evaluate the potential effect of radiofrequency ablation and deep brain stimulation in patients with treatment-refractory Tourette syndrome (TS), this study enrolled thirteen patients with TS who were admitted to our hospital between August 2002 and September 2018. Four patients received a single- or multi-target radiofrequency ablation after local, potentiated, or general anesthesia; eight patients underwent deep brain stimulation (DBS) surgery; and one patient underwent both ablation and DBS surgery. The severity of tics and obsessive compulsive disorder symptoms and the quality of life were evaluated using the Yale Global Tic Severity Scale (YGTSS), Yale–Brown Obsessive Compulsive Scale (YBOCS), and Gilles de la Tourette Syndrome Quality of Life scale (GTS-QOL), respectively, before surgery, one month after surgery, and at the final follow-up after surgery, which was conducted in December 2018. A paired-sample *t* test and a multiple linear regression analysis were performed to analyze the data. All patients underwent the operation successfully without any severe complications. Overall, the YGTSS total scores at one month post-surgery (44.1 ± 22.3) and at the final visit (35.1 ± 23.7) were significantly decreased compared with those at baseline (75.1 ± 6.2; both *p* < 0.05). Additionally, the YBOCS scores at one month post-surgery (16.5 ± 10.1) and at the final visit (12.0 ± 9.5) were significantly decreased compared with those at baseline (22.5 ± 13.1; both *p* < 0.05). Furthermore, the GTS-QOL scores at one month post-surgery (44.0 ± 12.8) and at the final visit (31.0 ± 17.8) were significantly decreased compared with those at baseline (58.4 ± 14.2; both *p* < 0.05). Results from a multiple linear regression analysis revealed that the improvement in the YGTSS total score was independently associated with the improvement in the GTS-QOL score at one month post-surgery (standardized β = 0.716, *p* = 0.023) and at the final visit (standardized β = 1.064, *p* = 0.000). Conversely, changes in YBOCS scores did not correlate with changes in GTS-QOL scores (*p >* 0.05). Our results demonstrate that tics, psychiatric symptoms, and the quality of life in patients with intractable TS may be relieved by stereotactic ablation surgery and deep brain stimulation. Furthermore, it appears that the improvement in tics contributes more to the post-operative quality of life of patients than does the improvement in obsessive compulsive symptoms.

## 1. Introduction

Tourette syndrome (TS) is a hereditary neurodevelopmental disorder with a typical onset in childhood, and its symptom severity shows prominent peak around the age of 11. The main clinical manifestations are sudden, rapid, recurrent, irregular, nonrhythmic motor, vocal, or phonic tics that are often accompanied by psychiatric comorbidities. The genetic basis of TS is not well elucidated and the underlying mutations remain elusive, except for rare cases. Accumulating evidence suggests that functional alterations in the basal ganglia play a role in the pathophysiology of TS, but the mechanisms remain unclear [1]. Approximately 50% of TS patients achieve complete or nearly complete tic remission, 30–50% experience significantly reduced symptom severity, and 5–10% of TS patients present sustained or worsened symptoms [2]. The therapeutic strategies for TS include habit reversal training and cognitive behavioral therapy as the first-line treatments; pharmacotherapy, including antipsychotic agents, as the second-line treatment if patients fail to respond to a behavioral intervention; and deep brain stimulation (DBS) as the third-line treatment [3].

Although DBS surgery is the last-choice treatment for intractable TS, previous studies have demonstrated that it has the potential to ameliorate tics. A recent study that enrolled 185 patients from 10 countries into the International DBS Database and Registry revealed that the mean Yale Global Tic Severity Scale (YGTSS) score improved by 45.1% one year after DBS implantation [4]. However, the targeted brain region for DBS remains critical and debatable. The current stage of clinical evidence suggests that the thalamus and globus pallidus internus (GPi) are the most promising targets [5]; however, it is still difficult to choose between these two targets. In addition to DBS, ablative surgery, including capsulotomy and pallidotomy, has been applied to treat TS. The outcomes of ablative surgery have demonstrated that it is effective and does not lead to severe or permanent side effects in patients [6,7]. Additionally, ablative surgery is advantageous compared to DBS because it does not require the implantation of a surgical device. However, DBS has almost completely replaced the ablative surgery due to the reversibility and adjustability of stimulation.

In our hospital, thirteen patients with treatment-refractory TS underwent stereotactic radiofrequency ablation or electrode implantation under local, potentiated, or general anesthesia. The aims of this study are mainly to assess the effectiveness and adverse events of stereotactic surgeries in TS patients. In this study, we comprehensively described the cases of patients who received ablative surgery over a seven-year follow-up period, a DBS case of an 11-year-old patient, a case of a patient who received DBS and ablative surgery simultaneously, and several cases of DBS targeting the GPi or the thalamus. We also conducted comparative and correlational analyses.

## 2. Patients and Methods

### 2.1. Patient Characteristics

We retrospectively assessed the long-term clinical outcomes and comorbid psychiatric disorders of thirteen patients (nine male) with intractable TS. These patients were admitted to our hospital for surgical treatment between August 2002 and September 2018. The four patients who were initially admitted underwent ablative surgery. The remaining nine patients underwent DBS surgery, and one of these patients received ablative surgery and DBS simultaneously because he experienced severe psychiatric symptoms. The age range of the patients was 11–32 years (mean age: 22.4 ± 7.2 years), and the age at symptom onset ranged from 5–29 years (mean time: 10.5 ± 6.2 years). Most patients presented with a comorbidity, including obsessive compulsive disorder (OCD; 10 cases), attention-deficit/hyperactivity disorder (2 cases), emotional disorder (12 cases), and self-injury behavior (3 cases). The demographics and clinical characteristics of the patients are detailed in Table 1. All patients met the diagnostic criteria for TS according to the Diagnostic and Statistical Manual of Mental Disorders, 5th Edition (DSM-5) and exhibited complex motor tics that were complicated by phonic tics. The inclusion and exclusion criteria of the study are as follows. Inclusion criteria: Severe tics, as defined by YGTSS >25/50; Tics are a primary cause of disability; Tics are refractory to conservative therapy; Psychosocial environment is stable; Demonstrated ability to adhere to recommended treatments; Neuropsychological profile indicates candidate can tolerate demands of surgery, postoperative follow-up, and possibility of poor outcome. Exclusion criteria: Active suicidal or homicidal ideation within 6 months; Active or recent substance abuse; Structural lesions on brain MRI; Malingering, factitious disorder, or psychogenic tics. The study was performed in accordance with the Declaration of Helsinki and Good Clinical Practice guidelines, and it was approved by the Ethics Committee of Tangdu Hospital (ethic approval code: TDLL-202205-03). All patients or their legal representatives provided written informed consent before enrolment.

### 2.2. Surgical Procedures

All the surgeries were performed by the same surgeon. We used a CRW (Radionics Inc., Burlington, MA, USA) or Leksell (Elekta AB, Stockholm, Sweden) stereotactic system and planning software. First, we performed a magnetic resonance imaging (MRI) scan (1.5 Tesla, GE Signa HDxt/3.0 Tesla, GE Signa EXCITE, Chicago, IL, USA), computed tomography (CT) scan with a 64-row multidetector scanner (GE LightSpeed VCT) after frame placement, or co-registration of pre-op MRI and stereotactic MRI or CT. Then, we identified the targets according to their common three-dimensional coordinates and the Schaltenbrand-Wahren atlas. These targets included the GPi, the centromedian–parafascicular complex (CM-Pf), the ventralis intermedius nucleus of the thalamus (Vim), the nucleus accumbens (NAc), the anterior limb of the internal capsule (ALIC), the amygdala, and the cingulate gyrus.

Two patients in our study population required local and potentiated anesthesia, whereas the other eleven patients required general anesthesia for the stereotactic surgery. Twelve of the patients received straight or semicircular incisions on their bilateral frontoparietal scalps, and one patient underwent ablative surgery with a unilateral straight incision. During the operation of the first seven patients, holes were drilled into the skull at the following coordinates: 1.0 cm anterior to the coronal suture and 3.5 cm lateral to the midline; these locations were similar to the positions marked using trajectory planning in the six remaining patients. After the hole was drilled into the skull, we ensured that the dura mater had been coagulated before opening it via a cruciate incision. The subsequent procedures differed among patients. One patient underwent ablative surgery targeting the left Vim, three patients received multi-target radiofrequency thermolesions, eight patients had an impulse generator placed under the skin and electrodes that targeted the GPi or the CM-Pf implanted into the brain, and one patient received an ablation followed by DBS surgery during the same operation. The surgical methods for the thirteen patients are presented in more detail in Table 2, whereas the coordinates of the DBS targets are described in Table 3.

The thermolesions were created at 80 °C for 60 s using a radiofrequency generator (Universal RF System, Radionics; or Leksell Neuro Generator, Elekta). The exposed part of the monopolar lesion electrode was 1.6 mm × 5 mm or 2 mm × 4 mm (diameter × length). The implanted leads were Model 3387 or 3389 (Medtronic, Minneapolis, MN, USA), Model 1210 (SceneRay, Suzhou, China), or Model L301 (Pins, Beijing, China). The leads were intracranially implanted and fixed in place in two patients, and the clinical outcomes and adverse effects were observed for several days to help us determine the next surgical procedures. After the temporary stimulation period in these two patients and the lead implantation surgery in the remaining six DBS patients, we incised the retroauricular and subclavian regions and implanted the connecting wire and implantable pulse generator (IPG; Model Kinetra 7428, PC 37601, or RC 37612 by Medtronic, Model 1180 by SceneRay, or Model G102R by Pins). The DBS devices were switched on one month after surgery. The stimulation parameters mainly consisted of a unipolar stimulation mode (one contact or two contacts as cathodes), a pulse width of 60–90 μs, a frequency of 80–180 Hz, and an amplitude of 2.6–3.5 V. The stimulation parameters were individually adjusted according to the extent of the symptomatic improvement and the degree of side effects with the goal of obtaining an optimal treatment with minimal side effects. The stimulation parameters at the final assessment are presented in Table 4. During the study period, patients continued the use of any medications that they were prescribed prior to surgery.

### 2.3. Efficacy Assessment

The follow-up data were compiled by an assessment group that was independent from the surgery group. The severity scores for motor tics, phonic tics, overall damage, and global tics were assessed using the YGTSS. The postoperative improvement rate of symptoms was calculated as follows:Improvement rate = (preoperative YGTSS total score − postoperative YGTSS total score)/preoperative YGTSS total score × 100%.

The symptoms of OCD were assessed using the Yale–Brown Obsessive Compulsive Scale (YBOCS), and quality of life was assessed using the Gilles de la Tourette Syndrome Quality of Life scale (GTS-QOL). These assessments were performed three days before surgery, one month after surgery, and during the last follow-up visit after surgery, which was conducted in December 2018.

### 2.4. Statistical Analysis

All statistical analyses were performed using SPSS version 22.0 (SPSS Inc.). Prior to statistical analysis, the Kolmogorov–Smirnov test and Shapiro–Wilk test were used to examine whether the values followed a normal distribution. Levene’s test was used to assess the homogeneity of variance. Data are presented as the mean ± standard deviation (SD). The primary outcome was the difference in the total YGTSS score between the preoperative (pre-op) and the postoperative (post-op) follow-up assessments. The secondary outcome was the difference in the YBOCS and GTS-QOL scores between the preoperative and the postoperative follow-up assessments. Specifically, the Friedman test followed by Dunn’s post hoc test and ANOVA followed by Tukey’s multiple comparisons test were used to determine whether there was a significant difference between the scores at baseline and at the two follow-up assessments. A multiple linear regression model was used to evaluate whether the improvement in the total YGTSS score or YBOCS score was related to the improvement in GTS-QOL score. These data were analyzed using a one-sample Kolmogorov–Smirnov test and homogeneity variances test. The correlation between the independent and dependent variables is expressed as an R-value. For all analyses, *p* < 0.05 was considered statistically significant.

## 3. Results

### 3.1. Postoperative Brain Images

The post-op MRI, CT, and fused images of the pre-op MRI and post-op CT helped to confirm the location of the radiofrequency lesions and the implanted leads. The accuracy of the lesions and the lead placements were verified in the thirteen patients. Figure 1 demonstrates the location of the leads and lesions on post-op MRIs and CTs of Cases 6–8 and 13. The locations of the DBS leads on post-op imaging data of Cases 9–12 are presented in Appendix A.

### 3.2. Tic Assessment

The four patients who only received thermolesions reported a clinical benefit regarding their tics within one week after the ablation. Additionally, the eight patients who only received DBS exhibited varying degrees of symptom amelioration during their temporary stimulation period or after the activation of their IPGs, and the one patient who received both ablation and DBS reported a benefit regarding tics within one week after the surgery and exhibited a further improvement in tics after his IPG was activated. Among the patients who only received DBS surgery, one patient’s tic symptoms were partially relieved three months after DBS surgery, but the long-term effect was unsatisfactory (Case 6). Another patient’s tic symptoms completely vanished after DBS surgery; however, he relapsed at the one-year follow-up because the IPG was unexpectedly switched off. The clinical improvement returned as soon as the IPG was reactivated (Case 5).

Detailed score data were obtained before surgery, one month post-surgery, and at the last follow-up. The total YGTSS scores of all thirteen patients were significantly decreased one month after surgery (44.1 ± 22.3) and at the last follow-up (35.1 ± 23.7) compared with those at baseline (75.1 ± 6.2; both *p* < 0.05) (Figure 2A). Additionally, motor tics, phonic tics, social impairment, and global scores were improved by 50.0%, 55.8%, 54.0%, and 53.3%, respectively, at the last follow-up compared with those at baseline.

### 3.3. Evaluation of Obsessive Compulsive Disorder

The YBOCS scores of eleven patients were significantly improved at one month post-surgery (16.5 ± 10.1) and at the last follow-up visit (12.0 ± 9.5) compared with those at baseline (22.5 ± 13.1; both *p* < 0.05) (Figure 2B). Therefore, the OCD severity was relieved after the operation.

### 3.4. Quality of Life Assessment

We used the GTS-QOL scale to assess the quality of life in ten of the TS patients who were later admitted to our hospital. Overall, we found that the GTS-QOL scores of the patients one month after surgery (44.0 ± 12.8) and at the last follow-up assessment (31.0 ± 17.8) were significantly improved compared with those at baseline (58.4 ± 14.2; both *p* < 0.05) (Figure 2C).

The above clinical outcomes are presented in Table 5.

### 3.5. Correlations between Quality of Life and the Severity of Tics and OCD

Results from the linear regression analysis demonstrated that the improvement in YGTSS total score was a significant independent predictor that correlated with the improvement in the GTS-QOL score one month after surgery (standardized β = 0.716, *p* = 0.023) and at the final follow-up visit (standardized β = 1.064, *p* = 0.000); However, the improvement in YBOCS scores did not correlate with the improvement in GTS-QOL scores (*p* > 0.05). A positive change in the YGTSS score indicated improved quality of life after stereotactic surgery. The correlations between the changes in GTS-QOL scores and the changes in YGTSS scores are shown in Figure 3. The contributions of the independent variables to the improvement in GTS-QOL score are shown in Table 6. The beta values indicated that these independent variables contributed strongly to the improvement in GTS-QOL scores. Table 7 exhibits the R-values (0.795 and 0.948).

### 3.6. Complications

Two patients (Case 2 and Case 3) who received ablative surgery that involved the bilateral cingulate gyrus, amygdalae, and ALIC presented apathy, urinary incontinence, and stereotypic movements. Of these two patients, one patient (Case 2) also displayed hoarseness, dysphagia, and reduced strength in expectoration and pronunciation. Another patient (Case 1) experienced a mild temporary contralateral hemiparesis after ablation of the left Vim. One patient (Case 4) experienced transient agitation, mild hypomnesis and confusion, mild memory loss, and right-handed weakness after the ablation of the bilateral amygdalae, NAc, ALIC, and left GPi. This patient’s post-op CT displayed an obvious focal edema that surrounded the lesions. One patient (Case 8) was temporarily affected by dysphoria, dizziness, and a mild motivation and memory impairment after bilateral NAc and ALIC ablation combined with bilateral GPi stimulation surgery. Of the above complications, apathy and memory impairment persisted, even if mildly. There were no severe side effects in the other eight patients with implanted electrodes; however, some reversible stimulation-related adverse events occurred. For instance, the patient with leads that were placed in CM-Pf (Case 6) had suffered from dizziness, blurred vision, lethargy, and numbness of lips and oral cavity when he received programming. The patients’ complications are presented in more detail in Table 2.

## 4. Discussion

All enrolled patients with intractable TS responded to surgical ablation or deep brain stimulation treatment with a reduction in symptoms and an improvement in the quality of life. These results are similar to previous studies [6,8]; therefore, our study has contributed to further support for the selection between DBS and ablation therapy for the treatment of TS.

Two reviews [9,10] described the inclusion and exclusion criteria for DBS in TS patients, and this included an age limit of 18 years; however, this limit was not an absolute criterion if a case was considered “urgent” by the local ethics committee. The suggested tic severity was defined as a YGTSS score above 35/50. According to these guidelines, tics had to be the primary cause of disability, and they had to be refractory to conservative therapy. In our study, we removed the suggested age limit of 18 years and enrolled an 11-year-old male patient (Case 5) for DBS surgery. We hold the opinion that a suitable age needs to be clearly defined in accordance with the patient’s characteristics. Recently, arguments for and against DBS in children and adolescents with TS have been published [11], and more stress has been placed on the potential long-lasting harmful effects of the disorder; therefore, DBS surgery has been considered for younger TS patients [12]. For example, Huasen et al. presented the case of a 12-year-old male who received DBS surgery in a clinical study [13]. Additionally, the morphological development of the brain of some adolescent TS patients is complete; therefore, the displacement of the implanted electrodes that is observed in younger patients is prevented in these patients. Although adolescent TS patients have a higher probability of spontaneous recovery, the persistent symptoms tend to severely impair their psychological development and social communication. This is especially observed in patients who are 10–12 years of age [14]. Accordingly, DBS in the pediatric population may be an effective option with a moderate safety profile for the treatment of TS in carefully selected children [15]. In this study, clinical attention was paid to weighing the risks and benefits, and ethical issues were fully discussed and cautiously considered before the surgery for Case 5.

Historically, the frontal lobe, thalamus, cerebellum, limbic system, and other regions were targeted by TS surgery [16]. The neurosurgical procedures for TS included bimedial frontal leucotomy, bilateral anterior cingulotomy, bilateral coagulation of the rostral intralaminar and medial nuclei of the thalamus, bilateral cerebellar dentatotomy, and bilateral anterior cingulotomy with infrathalamic lesions. In China, capsulotomy or pallidotomy was applied to treat severe TS, and the efficacy and safety of these procedures have been verified in several studies [6,7]. It remains unclear whether ablative surgery is a suitable primary or alternative strategy for TS patients, and whether the electrode implantation is a more appropriate option. According to our findings, we propose that the following factors should be used as a reference for radiofrequency ablation: patients who are at least 18 years old; the patient’s socioeconomic conditions conflict with DBS management; unilateral limb tics are increased in severity; patients with psychiatric comorbidities, including severe mania or hallucination; and the patient can accept transient unwanted effects. By contrast, the suggested reference factors for electrode implantation are as follows: patients who are younger than 18 years old (make sure the patient’s skull size does not differ substantially from that of an adult in order to avoid dislocation of the implanted lead after the complete development of his brain), have normal cognition and communication skills, and who are able to regularly visit a healthcare professional for programming. The wide application of DBS for movement disorders and its reversibility and adjustability have gradually made the lead placement the primary treatment choice for patients with TS. Nevertheless, the radiofrequency ablation should not completely be abandoned. Although bilateral surgery, regardless of lesion or neurostimulation, was generally deemed appropriate for the patients with severe bilateral disabilities [17], simultaneous bilateral lesions to the Vim caused aphasia, cognitive decline, and balance disturbance. Additionally, clinical reports have reported that abulia, anhedonia, apathy, and worsening of speech and memory were associated with bilateral pallidotomy [18,19]. Hence, we performed unilateral ablation of the thalamus or globus pallidus in Cases 1, 2, and 4 and believe that a unilateral lesion to the Vim or GPi may help alleviate contralateral predominant tics without any severe long-term complications in these patients. Another patient (Case 8) underwent ablative surgery with the NAc and ALIC as targets in addition to DBS surgery because his OCD symptoms and destructive behaviors were thought to be severe and treatment-refractory.

There are also several issues regarding DBS in TS patients that need to be addressed, including the objective demonstration of its efficacy for different aspects of TS, factors that predict the individual responsiveness, methods that derive the optimal stimulation parameters, and optimal choice of the stimulation target [20]. Among these issues, the selection of the best target is the most prominent. For that reason, DBS is still viewed as an experimental approach in TS patients. Based on the knowledge of TS pathophysiology, the identified target system is composed of the GPi, CM-Pf, NAc, ALIC, subthalamic nucleus (STN), and globus pallidus externa, which are all located in the cortico-striatal-thalamocortical circuit (CSTC). This circuit is related to movement, motivation, decision, execution, emotion, and memory. The classical models of the CSTC provide frameworks to uncover the neurophysiological basis for the manifestations of involuntary tics [2,21].

At least four randomized, double-blind, controlled clinical trials indicated that bilateral thalamic DBS effectively treated TS [22,23,24,25]. Specifically, Welter et al. found that bilateral stimulation of the GPi or CM-Pf resulted in a 65–96% and 30–64% reduction in tic severity, respectively, according to the YGTSS. The association of pallidal and thalamic stimulations did not lead to a further reduction in tic severity (43–76%) [26]. Additionally, Houeto et al. reported that bilateral CM-Pf DBS decreased tic severity by 65% on the YGTSS. Moreover, mood, anxiety, and impulsivity were improved. Bilateral GPi DBS also led to an improvement in tic severity (65% reduction) and self-injury behavior; however, mood and impulsivity were worsened following bilateral GPi DBS compared with mood and impulsivity following CM-Pf DBS. The association of thalamic and pallidal stimulation prevented self-injury behavior and reduced tic severity by 70% [27]. Shields et al. described a female patient with medically intractable TS whose symptoms only significantly improved after the bilateral placement of deep brain stimulators in the anterior inferior internal capsule. However, her symptomatic improvement was not complete, and only a 23% reduction in the global severity of YGTSS was observed. Thus, the electrodes were subsequently transferred into the centromedian nucleus of the thalamus, and this led to a significant improvement in tic control. Specifically, the global severity of YGTSS was decreased by 46% three months after the second surgery compared with that at baseline [28]. Furthermore, results from a study conducted by Kakusa B et al. revealed that bilateral, dual-target DBS to the CM and ventral capsule/ventral striatum (VC/VS) effectively treated motor and non-motor complications of severe, intractable TS and its comorbidities, including OCD, major depressive disorder, chronic pain, and substance abuse [29]. Case studies regarding NAc stimulation with or without ALIC involvement have also been reported. Results from these studies demonstrated that the tic symptoms were relieved to different extents in all patients, and the obsessive compulsive symptoms were controlled in most patients [30]. Martinez-Torres et al. described a patient with Parkinson’s disease (PD) who also had a history of TS and underwent bilateral STN DBS. Following bilateral STN DBS, the patient experienced an improvement in both PD symptoms and tics. Specifically, STN DBS produced a 57% improvement in the motor part of the Unified Parkinson’s Disease Rating Scale and a 97% reduction in tic frequency at the one-year follow-up [31]. Furthermore, a systematic review and a meta-analysis revealed a significant reduction in YGTSS scores after the stimulation of the thalamus, posteroventrolateral, or anteromedial part of the GPi, ALIC, and NAc with no significant differences between these targets [32].

In the current study, we performed CM-Pf DBS on one TS patient (Case 6). This patient’s tics were partially relieved three months after the operation, but the long-term effect was unsatisfactory. However, patients who underwent GPi DBS achieved relatively good outcomes. This also included the 11-year-old patient (Case 5) whose tics and obsessive compulsive symptoms improved after the operation. Although this patient relapsed at the one-year follow-up because the IPG was accidentally switched off, his tics disappeared as soon as the IPG was reactivated. Similarly, Kimura Y et al. reported that two patients experienced a relapse in tics after the termination of DBS, but they were responsive to the reintroduction of DBS [33].

The thalamus and the globus pallidus are currently the most popular and useful targets for DBS in patients with TS. Since both targets results in a similar efficacy, it is difficult to preferentially select one over the other. However, decisions can also be made according to the target-specific adverse effects and the programming parameters, which means power consumption and battery life. Different target stimulation may result in different adverse effects. For example, stimulation of the thalamus may lead to sedation, fatigue, reduced energy, apathy, lethargy, decrease in sexual functions, or transient blurring of vision, while the stimulation of the GPi may cause nausea, vertigo, anxiety, or short dystonic jerks [34]. Therefore, we can avoid choosing some targets for DBS according to the patient’s preoperative status and symptoms. Results from a literature review revealed that the optimized stimulation parameters for thalamic stimulation are as follows: frequency, between 65 and 185 Hz; pulse width, between 60 and 210 s; and amplitude, between 2 and 8.5 V. In contrast, the optimized stimulation parameters for GPi DBS are as follows: frequency, 20–185 Hz; pulse width, 60–270 s; and amplitude, 2–5 V. The configuration is often set as a unipolar mode (one contact or two contacts as cathodes) or bipolar mode [35].

The CM-Pf is the most frequently used surgical target for treating TS. CM-Pf DBS has a sustained effect on tics, behavior, quality of life, and drug demand, and its efficacy for motor tics is superior to that for vocal tics. In China, the GPi is widely employed because it is easily identified and located in MRI images prior to lead placement. Additionally, the treatment of dystonia provided surgeons with a lot of experience for targeting the GPi. Findings from a few small-sample studies reported that GPi DBS was more effective than thalamus stimulation for tic control. However, the difference between its posteroventrolateral part (marginal region) and anteromedial part (sensorimotor region) remains unclear. An image-based analysis illustrated that the regions within, superior, or medial to the GPi were associated with a greater improvement in obsessive compulsive behavior than regions that were inferior to the GPi [36]. DBS in the ALIC-NAc is seldom applied and tends to contribute to an improvement in obsessive compulsive symptoms and psychosocial functions apart from alleviating tics. Despite the lack of large-sample randomized controlled trials to confirm the most effective DBS target, we believe that the GPi is the optimal target for DBS surgery in TS patients based on the findings from case reports and our clinical experience.

The previous study revealed that the improvement in tics seemed to be positively correlated with improved functional outcome, and symptomatic improvement might lead to unexpected major psychosocial changes [37]. GTS-QOL was reported to be most strongly associated with the scores from rating scales for some psychiatric symptoms, and YGTSS total tic score [38]. In this study, we found the improvement in the YGTSS total score correlated with an improvement in the GTS-QOL score after operation, indicating that the relief of tics in patients significantly promoted their overall life state.

This study also had some limitations. For example, our sample size was not large enough to compare differences in the efficacy between ablation and stimulation or between the different targets of the stereotactic surgery. Additionally, among the 13 patients in this study, only 3 were female patients. The effects of surgical treatment on these three patients were satisfactory. The improvement rates of the total YGTSS scores were all more than 60%, and the YBOCS and GTS-QOL scores were also significantly improved. Unlike male patients, there was no female patient with the poor curative effect. However, due to the small number of cases, gender differences in terms of recovery after surgical treatment for TS have not been found.

## 5. Conclusions

Results from our study demonstrate that tics, psychiatric symptoms, and quality of life in patients with intractable TS may be relieved by stereotactic ablation surgery and DBS. However, it appears that the improvement in tics contributes more to the postoperative quality of life of patients than does the improvement in obsessive compulsive symptoms. Additionally, attention should be paid to the risk-benefit balance in adolescent patients who may experience spontaneous tic remission. Furthermore, ablative surgery alone or in combination with DBS surgery may be used to effectively treat some patients. Both the inclusion criteria and the details of the surgical procedure should be taken into account before the intervention.

## Figures and Tables

**Figure 1 brainsci-12-00838-f001:**
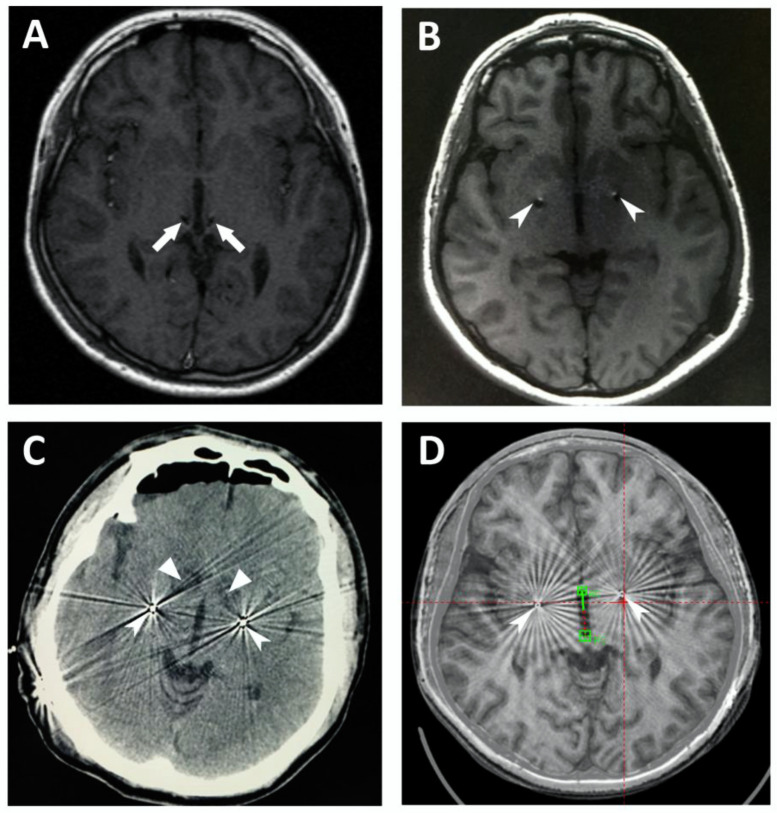
Location of the leads and lesions on postoperative MRIs and CTs. (**A**) Case 6, magnetic resonance imaging (MRI) scan of the brain after lead implantation, demonstrating the bilateral leads in the centromedian–parafascicular complex (CM-Pf); (**B**) Case 7, brain MRI scan after lead implantation, showing the bilateral leads in the globus pallidus internus (GPi); (**C**) Case 8, computed tomography (CT) scan of the brain after ablation and lead implantation, demonstrating the bilateral lesions in the anterior limb of the internal capsule (ALIC) and the bilateral leads in the GPi; (**D**) Case 13, fusion of the postoperative CT with preoperative MRI, demonstrating the bilateral leads in the GPi. The arrows denote CM-Pf, the arrowheads denote GPi, and the triangles denote ALIC.

**Figure 2 brainsci-12-00838-f002:**
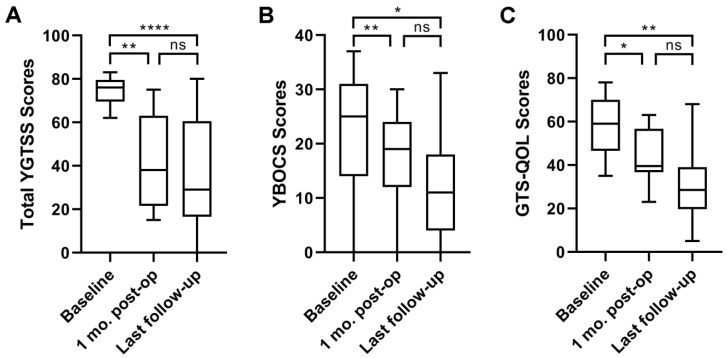
The assessment of tics, obsessive compulsive disorder, and quality of life of the TS patients. The total YGTSS scores (**A**), YBOCS scores (**B**), and GTS-QOL scores (**C**) of the patients one month after surgery and at the last follow-up assessment were significantly decreased compared with those at baseline (A: *p* < 0.05, Friedman test followed by Dunn’s post hoc test, *n* = 13, number of patients for A assessment; B, C: *p* < 0.05, ANOVA followed by Tukey’s multiple comparisons test, *n* = 11/10, number of patients for B/C assessment). Data are presented in boxplots. * *p* < 0.05, ** *p* < 0.01, and **** *p* < 0.0001; ns, no significant difference.

**Figure 3 brainsci-12-00838-f003:**
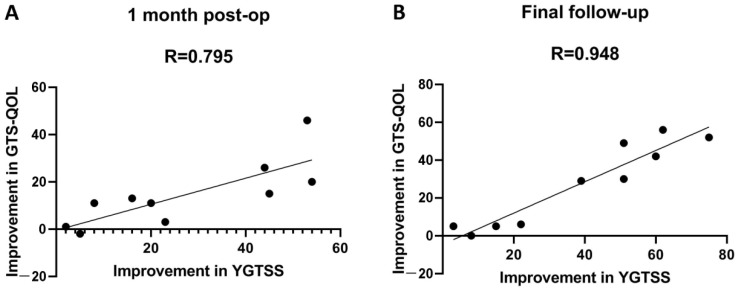
Scatter diagram of the correlation between the decreased score of GTS-QOL and YGTSS. Improvement in YGTSS total score (x-axis) correlates with improvement in GTS-QOL score (y-axis) both one month after operation (**A**) and at the final follow-up (**B**). Each black dot represents a case (4–13).

**Table 1 brainsci-12-00838-t001:** Baseline clinical characteristics of thirteen patients with TS.

Patient	Sex	Age at Symptom Onset (Years)	Age at Surgery (Years)	Core Symptoms	Comorbid Neuropsychiatric Disorders	Medication Before Surgery	Follow-Up (Months)
1	Male	12	19	Facial grimaces; Eye blinking; Right arm jerks; Neck rotation; Shriek	OCD; Mania; Cognitive impairment	Haloperidol; Benzhexol	12, and then lost in follow-up
2	Male	5	15	Eye blinking; Pout; Right arm jerks; Head-striking; Phonate	Mild cognitive impairment; Destructive behavior	Haloperidol; Benzhexol; Clonazepam	143
3	Male	14	21	Head hypsokinesis; Right arm elevation; Coprolalia; Bark	OCD; Irritability; Destructive behavior	Haloperidol; Benzhexol; Aripiprazole; Sulpiride; Clozapine	139
4	Female	9	22	Eye blinking; Claw at her face; Shoulder shrugs; Shake head; Stamp feet; Right limb jerks; Throat-clearing; Shriek; Coprolalia	Depression; Mania; Self-injury behavior	Haloperidol	83
5	Male	8	11	Nod head; Shoulder shrugs; Limb jerks; Hiccup sound	OCD	Chinese medicine	82
6	Male	5	14	Eye blinking; Throat-clearing; Spitting ptysis; Twist neck; Squat; Hand jerks; Shouting; Coprolalia	ADHD; Depression; Anxiety; OCD; Memory deficits; Auditory hallucination; Delusion; Irritability	Haloperidol; Aripiprazole; Risperidone; Benzhexol; Clonazepam; Sertraline; Atomoxetine; Olanzapine; Topiramate	71
7	Female	29	32	Eye blinking; Shoulder shrugs; Shake head; Stamp feet; Face hitting with right hand; Limb jerks; Shriek; Throat-clearing	ADHD; Anxiety; OCD; Depression; Self-injury behavior	Haloperidol; Benzhexol; Lorazepam	62
8	Male	7	23	Eye blinking; Hit face and neck; Stamp feet; Phonate	OCD; Irascibility; Destructive behavior	Unknown	46
9	Female	10	30	Eye blinking; Neck extension; Shoulder shrugs; Shriek	Anxiety; Depression; Emotional lability	Haloperidol; Tiapride	24
10	Female	13	27	Facial grimaces; Shoulder shrugs; Grit teeth; Phonate	Anxiety; Depression; OCD; Emotional lability	Haloperidol; Risperidone; Clonazepam; Lorazepam; Clozapine	19
11	Male	7	19	Limb jerks; Phonate; Coprolalia	Anxiety; Depression; OCD; Mania	Haloperidol; Faverin; Clonazepam; Tetrabenazine	15
12	Male	10	32	Eye blinking; Hit face and neck; Bite the quilt; Pull his hair and eyebrows; Phonate	Anxiety; OCD; Phobia; Irascibility; Destructive behavior	Haloperidol; Tiapride; Aripiprazole	12
13	Male	8	13	Eye blinking; Shoulder shrugs; Neck extension; Limb jerks; Stamp feet; Face hitting with hands; Throat-clearing; Shriek; Coprolalia	Anxiety; Depression; OCD; Self-injury behavior; Mania	Tiapride; Haloperidol; Aripiprazole; Clonazepam; Benzhexol; Risperidone; Eperisone	3

TS, Tourette syndrome; OCD, obsessive compulsive disorder; ADHD, attention-deficit/hyperactivity disorder.

**Table 2 brainsci-12-00838-t002:** Details of the surgical procedures in the thirteen TS patients.

Patient	Surgery	Anesthesia	Targets	Images Used for Targeting	Complications
1	Ablation	Local	Left Vim	CT	Short-term mild contralateral hemiparesis
2	Ablation	Local and potentiated anesthesia	Bilateral amygdale, ALIC, cingulate gyrus, and left GPi	CT	Short-term apathy, urinary incontinence, stereotypic movement, hoarseness, and dysphagia
3	Ablation	General	Bilateral amygdale, ALIC, and cingulate gyrus	MRI	Temporary headache, hyperpyrexia, and short-term urinary incontinence
4	Ablation	General	Bilateral amygdale, NAc, ALIC, and left GPi	MRI	Short-term agitation, mild hypomnesis and confusion, mild memory impairment, and right-hand weakness
5	DBS	General	Bilateral GPi	MRI	None
6	DBS	General	Bilateral CM-Pf	MRI	None
7	DBS	General	Bilateral GPi	MRI	None
8	Ablation/DBS	General	Bilateral NAc and ALIC/bilateral GPi	MRI	Temporary dysphoria and dizziness, short-term mild motivation, and memory impairment
9	DBS	General	Bilateral GPi	MRI	None
10	DBS	General	Bilateral GPi	Co-registration of pre-op MRI and stereotactic CT	None
11	DBS	General	Bilateral GPi	MRI	None
12	DBS	General	Bilateral GPi	Co-registration of pre-op MRI and stereotactic CT	None
13	DBS	General	Bilateral GPi	Co-registration of pre-op MRI and stereotactic MRI	None

TS, Tourette syndrome; Vim, ventralis intermedius nucleus of the thalamus; ALIC, anterior limb of the internal capsule; GPi, globus pallidus internus; NAc, nucleus accumbens; CM-Pf, centromedian–parafascicular complex.

**Table 3 brainsci-12-00838-t003:** Coordinates of the DBS targets.

Patient	Manufacturer	Lead Model	AC-PC Length (mm)	Left			Right		
				X	Y	Z	X	Y	Z
5	Medtronic	3387	22.1	18.5	2.0	−5.0	19.5	2.5	−4.5
6	Medtronic	3387	24.0	7.0	−8.0	0.0	7.0	−8.0	0.0
7	Medtronic	3387	22.0	19.0	3.5	−4.5	20.0	3.0	−4.0
8	Medtronic	3387	20.9	21.5	2.5	−5.0	24.0	2.5	−5.0
9	SceneRay	1210	22.2	18.0	3.5	−4.0	18.0	3.5	−4.0
10	SceneRay	1210	22.8	20.0	3.0	−7.0	19.5	3.0	−6.0
11	Medtronic	3389	22.5	21.0	2.5	−4.0	20.5	3.0	−3.5
12	SceneRay	1210	22.6	21.0	4.5	−4.0	21.5	4.0	−3.5
13	Pins	L302	23.4	21.0	4.0	−4.0	21.5	4.0	−3.5

AC, anterior commissure; PC, posterior commissure; X = mm lateral of AC-PC; Y = mm anterior of mid AC-PC; Z = mm inferior to AC-PC.

**Table 4 brainsci-12-00838-t004:** Stimulation parameters at the final assessment.

Patient	Left Electrode					Voltage (V)	Pulse (µs)	Frequency (Hz)	Right Electrode					Voltage (V)	Pulse (µs)	Frequency (Hz)
	Case	0	1	2	3				Case	4/8	5/9	6/10	7/11			
5	+		-			2.6	60	130	+		-			2.6	60	130
6	+	-	-			3.1	90	145	+		-	-		3.3	90	145
7	+	-				3.4	60	160	+		-			3.4	60	160
8			-		+	2.7	60	150			-		+	2.7	60	150
9	+			-		3.0	90	180	+			-		3.0	90	180
10	+	-	-			3.0	90	130	+		-			3.0	90	130
11	+	-				2.9	60	130	+	-				2.9	60	130
12	+	-				3.0	60	80	+	-				2.85	60	80
13	+				-	2.85	100	130	+				-	2.85	100	130

**Table 5 brainsci-12-00838-t005:** Primary and secondary outcomes.

Patient	YGTSS Pre-op				YGTSS Post-op								Total YGTSS Scores (%)		YBOCS Pre-op	YBOCS Post-op		GTS-QOL Pre-op	GTS-QOL Post-op	
	Motor	Vocal	Impairment	Total	Motor		Vocal		Impairment		Total									
					1 mo.	Latest	1 mo.	Latest	1 mo.	Latest	1 mo.	Latest	1 mo.	Latest		1 mo.	Latest		1 mo.	Latest
1	18	14	30	62	6	7 *	6	6 *	10	10 *	22	23 *	64.5	62.9	-	-	-	-	-	-
2	20	10	40	70	5	5	0	0	10	10	15	15	78.6	78.6	-	-	-	-	-	-
3	13	23	40	76	8	5	10	11	20	20	38	36	50.0	52.6	31	20	16	-	-	-
4	20	18	40	78	7	6	6	0	20	10	33	16	57.7	79.5	0	0	0	78	63	22
5	14	21	40	75	6	0	5	0	10	0	21	0	72.0	100.0	24	15	4	57	37	5
6	23	20	40	83	17	21	18	19	40	40	75	80	9.6	3.6	14	12	10	73	62	68
7	19	9	40	68	5	7	0	0	10	10	15	17	77.9	75.0	18	12	9	69	23	20
8	23	16	40	79	10	9	5	0	20	10	35	19	55.7	75.9	36	19	12	67	41	25
9	19	20	30	69	16	16	18	15	30	30	64	61	7.2	11.6	0	0	0	35	37	35
10	21	19	40	80	15	8	15	11	30	10	60	29	25.0	63.8	37	30	11	49	38	19
11	24	19	40	83	17	15	13	16	30	30	60	61	27.7	26.5	31	29	33	39	36	33
12	19	16	40	75	19	16	14	14	40	30	73	60	2.7	20.0	25	21	18	56	55	51
13	20	18	40	78	18	12	14	7	30	20	62	39	20.5	50.0	31	24	19	61	48	32
Mean (SD)	19.5 ± 3.2	17.2 ± 4.1	38.5 ± 3.8	75.1 ± 6.2	11.5 ± 5.6	9.8 ± 5.8	9.5 ± 6.3	7.6 ± 7.1	23.1 ± 11.1	17.7 ± 11.7	44.1 ± 22.3	35.1 ± 23.7	41.3	53.3	22.5 ± 13.1	16.5 ± 10.1	12.0 ± 9.5	58.4 ± 14.2	44.0 ± 12.8	31.0 ± 17.8
*p*											0.000	0.000				0.003	0.006		0.010	0.003

Pre-op, preoperative; Post-op, postoperative; YGTSS, Yale Global Tic Severity Scale; YBOCS, Yale–Brown Obsessive Compulsive Scale; GTS-QOL, Gilles de la Tourette Syndrome Quality of Life scale; Total YGTSS Scores (%), improvement rates of the total YGTSS scores; Motor, motor tic scores; Vocal, phonic tic scores; Impairment, social impairment scores; Total, total YGTSS scores; 1 mo., one-month follow-up; Latest, the last follow-up; *, data were collected in the 12-month follow-up period; -, data were not collected.

**Table 6 brainsci-12-00838-t006:** Coefficients of the multiple linear regressions for the improved score of GTS-QOL.

	Model	Unstandardized Coefficients		Standardized Coefficients	t	Sig.
		B	Std. Error	Beta		
1 mon. post-op	(Constant)	−1.615	5.330		−0.303	0.771
	Improvement in YGTSS	0.497	0.172	0.716	2.894	0.023
	Improvement in YBOCS	0.481	0.677	0.176	0.710	0.501
Final follow-up	(Constant)	−4.377	4.173		−1.049	0.329
	Improvement in YGTSS	0.929	0.116	1.064	8.030	0.000
	Improvement in YBOCS	−0.408	0.283	−0.191	−1.441	0.193

Statistically significant *p*-values were written in bold term.

**Table 7 brainsci-12-00838-t007:** Model summary of improvement in GTS-QOL score based on linear regression analyses.

	R	R Square	Adjusted R Square	Std. Error of the Estimate
1 mon. post-op	0.795 ^a^	0.632	0.586	9.02204
Final follow-up	0.948 ^a^	0.899	0.887	7.37777

^a^ Predictors: (Constant), Improvement in YGTSS total score.

## Data Availability

The authors confirm that the data supporting the findings of this study are available within the article and its Appendix A.

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
