# Peer review of "Stereotactic Surgery for Treating Intractable Tourette Syndrome: A Single-Center Pilot Study"

_brainsci, 2022, doi:10.3390/brainsci12070838_

Round 1

Reviewer 1 Report

Summary: This clinical study aims to test the efficacy of radiofrequency ablation and deep brain stimulation in patients with TS. Although the sample size is quite small, the findings from this study corroborated the previous literature. The mention of complications in the results section is encouraging. Although the procedure is deemed invasive, the authors showed a ~53% recovery (total YGTSS score) which is remarkable. The manuscript is well written, and I have a few minor comments below.

Comments and Suggestions:

Change the following sentence in abstract- “A paired-sample t test and a multiple linear regression analysis were performed to analyzed the data” to “…. to analyze the data.”

§In figure 1, please show the location of CM-Pf, GPi, ALIC using an arrowhead.

§  It is recommended to provide the imaging data for all the 13 subjects either in figure 1 or in a supplementary figure.

§  Please ensure that the headings in Table 5 is not distorted in the final format.

§  It will be easy to follow if total YGTSS scores are shown as box whisker plots in addition to the table.

§  Although authors used paired-sample t test to calculate the p-Value, it is not clear if the data follows normal distribution. Please verify the distribution of the data.

§  The sample size is quite small with very few data points involving female patients. Please discuss any potential sex differences seen in this study in terms of recovery.

Author Response

Dear Editors and Reviewers:

Thank you for your letter and for the reviewers’ comments about our manuscript titled “Stereotactic surgery for treating intractable Tourette syndrome: A single-center series”(brainsci-1738139). All the comments are very valuable, helped us revise and improve our manuscript, and guided our studies. We have carefully studied these comments and have made corrections, which we hope will meet with approval. The revised portions are highlighted in red in the revised manuscript. The main corrections in the paper and the responses to the reviewer’s comments are as follows:

Response to Reviewer 1 Comments

Point 1: Change the following sentence in abstract- “A paired-sample t test and a multiple linear regression analysis were performed to analyzed the data” to “…. to analyze the data.”

Response 1: Thank you for the revision suggestion. I have modified it and checked the English language in the manuscript.

Point 2: In figure 1, please show the location of CM-Pf, GPi, ALIC using an arrowhead.

Response 2: Thank you for your suggestion. I have made the different targets indicated. “The arrows denote CM-Pf, the arrowheads denote GPi, and the triangles denote ALIC.”

Point 3: It is recommended to provide the imaging data for all the 13 subjects either in figure 1 or in a supplementary figure.

Response 3: Thank you for your valuable suggestions. In addition to the imaging data of Nos 6 - 8, and 13 patients, we presented the postoperative imaging data of Nos 9 - 12 patients in a supplementary figure. These newly added four postoperative CT or MRI images all show the bilateral GPi as the location of DBS leads. However, we can not obtain the imaging data of the first five cases due to more than ten years after operation.

Point 4: Please ensure that the headings in Table 5 is not distorted in the final format.

Response 4: I have the adjusted the size of the headings in Table 5 to make them correct in the format.

Point 5: It will be easy to follow if total YGTSS scores are shown as box whisker plots in addition to the table.

Response 5: Thank you for your valuable suggestion. In the revised manuscript, the preoperative and postoperative total YGTSS scores, YBOCS scores, and GTS-QOL scores are shown as box whisker plots in Figure 2.

Point 6: Although authors used paired-sample t test to calculate the p-Value, it is not clear if the data follows normal distribution. Please verify the distribution of the data.

Response 6: Thank you for your valuable suggestion. Prior to statistical analysis, the Kolmogorov–Smirnov test and Shapiro–Wilk test were used to examine whether the values followed a normal distribution. Levene's test was used to assess the homogeneity of variance. Therefore, We have changed some statistical methods according to the homogeneity of variance of our total YGTSS scores data. In our analyses, Friedman test followed by Dunn’s post-hoc test and ANOVA followed by Tukey's multiple comparisons test were used for pairwise comparisons of multiple samples.

Point 7: The sample size is quite small with very few data points involving female patients. Please discuss any potential sex differences seen in this study in terms of recovery.

Response 7: Thank you for your valuable suggestion. We discussed the potential sex differences seen in this study in terms of recovery.

Reviewer 2 Report

In the manuscript entitled “Stereotactic surgery for treating intractable Tourette syndrome: A single-center series”, Wang and colleagues reported the treatment outcomes of a pilot study of thirteen patients with treatment-refractory Tourette syndrome (TS). The patients underwent stereotactic radiofrequency ablation and/or electrode implantation deep brain stimulation (DBS). I have the following recommendations for the authors.

l   Please revise the language with professional English editing. Some grammatical mistakes are detected. For example, in abstract, “A paired-sample t test and a multiple linear regression analysis were performed to analyzed the data.” It should be “analyze”.

l   Please insert “pilot” in the title to reflect the study category.

l   Please state the study aims clearly at the end of introduction section and elsewhere in abstract.

l   Please re-edit the words inside Table 5 to be comprehensive.

l   One of the main limitations, as stated by the authors, is the relatively small sample size. Please provide the power of your study. Studies with small to moderate samples size employing linear regression analysis may not be accurate. I am not a statistician, but the authors should discuss with this issue. Moreover, statistical methods should be reconfirmed by experts.

l   Apart from small sample size, other limitations may include:

1.       The inclusion and exclusion criteria of the study were not mentioned for clarity.

2.       Results are from single center, not from multiple centers.

3.       Seemed not blinded research, the patients know which surgery is chosen for them.

4.       Surgeries performed by different surgeons, not the same surgeon, at least not mentioned in the text.

5.       Surgical procedures not standardized, details of the procedures showed discrepancies among similar surgeries.

l   In discussion, the authors did not compare their approach with other state-of-the-art approaches.

l   Please demonstrate how this study adds value to the current literature regarding this topic.

Author Response

Dear Editors and Reviewers:

Thank you for your letter and for the reviewers’ comments about our manuscript titled “Stereotactic surgery for treating intractable Tourette syndrome: A single-center series”(brainsci-1738139). All the comments are very valuable, helped us revise and improve our manuscript, and guided our studies. We have carefully studied these comments and have made corrections, which we hope will meet with approval. The revised portions are highlighted in red in the revised manuscript. The main corrections in the paper and the responses to the reviewer’s comments are as follows:

Response to Reviewer 2 Comments

Point 1: Please revise the language with professional English editing. Some grammatical mistakes are detected. For example, in abstract, “A paired-sample t test and a multiple linear regression analysis were performed to analyzed the data.” It should be “analyze”.

Response 1: Thank you for your helpful suggestion. I have revised the language with professional English editing.

Point 2: Please insert “pilot” in the title to reflect the study category.

Response 2: Thank you for your valuable suggestions. I have inserted “pilot” in the title.

Point 3: Please state the study aims clearly at the end of introduction section and elsewhere in abstract.

Response 3: Thank you for your valuable suggestion. I have stated the study aims clearly at the end of introduction section. “The aims of this study are mainly to assess the effectiveness and adverse events of stereotactic surgeries in TS patients.”

Point 4: Please re-edit the words inside Table 5 to be comprehensive.

Response 4: Thank you for your modification suggestion. I have re-edited the words inside Table 5 to make them easier to be understood.

Point 5: One of the main limitations, as stated by the authors, is the relatively small sample size. Please provide the power of your study. Studies with small to moderate samples size employing linear regression analysis may not be accurate. I am not a statistician, but the authors should discuss with this issue. Moreover, statistical methods should be reconfirmed by experts.

Response 5: Thank you for raising this important issue. Due to the medical ethics problem and the low incidence of refractory TS, there are currently very few cases of TS patients who have received the stereotaxic surgery in the world. As a single-center report, the  sample size of 13 in our center is not a small one. In terms of statistical power, only 10 samples in this study probably lead to an inaccurate linear regression analysis. However, the linear regression analysis was performed with the same size of samples in a previous study (PMID: 16458260). After reevaluating our statistical data, based on their normal distribution and homogeneity of variance among groups, we employed the more appropriate statistical methods, ANOVA  and Friedman test instead of the paired t test.

Point 6: Apart from small sample size, other limitations may include: The inclusion and exclusion criteria of the study were not mentioned for clarity.

Response 6: We inserted the inclusion and exclusion criteria in “2.1. Patient characteristics”.

Point 7: Results are from single center, not from multiple centers.

Response 7: This is a single-center study, and the results are not from multiple centers.

Point 8:  Seemed not blinded research, the patients know which surgery is chosen for them.

Response 8: This is a retrospective study, and does not use a double-blind design.

Point 9: Surgeries performed by different surgeons, not the same surgeon, at least not mentioned in the text.

Response 9: All the surgeries were performed by the same surgeon, the corresponding author of the manuscript. It has been mentioned in the text.

Point 10: Surgical procedures not standardized, details of the procedures showed discrepancies among similar surgeries.

Response 10: Thank you for your valuable comments. The first four patients (Cases 1-4) were treated with radiofrequency ablations with the different targets from each other. However, they generally shared the bilateral amygdale, AlIC and cingulate gyrus as their common targets, which were used to relieve the psychiatric symptoms. Because they suffered some other special symptoms in addition to the basic TS symptoms, their ablative targets were not exactly the same. For example, in Case 1 and 4, owing to the severe right limb jerks, the left Vim or GPi was chosen for ablation in order to better control the involuntary movements of limbs, reflecting the individualization of treatment. The last nine patients were all treated with DBS surgeries, except for Case 8, who received the radiofrequency ablation due to severe psychiatric symptoms including OCD, irascibility, and destructive behaviors besides DBS surgeries. In Case 6, we performed the lead placement in CM-Pf, which was the most widely used in the world. As a result of poor outcomes of CM-Pf-DBS, bilateral GPi were chosen as the targets for DBS treatment for the remaining seven patients. The surgical procedures for these seven patients remained consistent.

Point 11: In discussion, the authors did not compare their approach with other state-of-the-art approaches.

Response 11: Thank you for your important suggestion. The therapeutic strategies for TS include habit reversal training and cognitive behavioral therapy as the first-line treatments; pharmacotherapy, including antipsychotic agents, as the second-line treatment; and deep brain stimulation (DBS) as the third-line or fourth-line treatment. Therefore, DBS is regarded as the last choice for refractory TS patients, and it is still the most advanced method to treat TS at present. No other therapy is listed as the third-line or fourth-line treatment. Additionally, it is meaningless to compare different lines of treatments.

Point 12: Please demonstrate how this study adds value to the current literature regarding this topic.

Response 12: There are two prominent issues regarding DBS in TS patients that need to be addressed, including factors that predict the individual responsiveness, and optimal choice of the stimulation target. The previous studies mainly focused on the inclusion of candidate patients and the selection of best DBS targets. Our study adds further support for the selective use of DBS or ablation therapy for the treatment of TS. Particularly, in our case series, we comprehensively reported the cases of patients who received radiofrequency ablation surgery over a seven-year follow-up period, a DBS case of a young (11-year-old) patient, a case of patient who received DBS and ablative surgery simultaneously, which were rarely reported in similar published literatures. Most importantly, our study has contributed to further support for the selection between DBS and ablation therapy for the treatment of TS.

Round 2

Reviewer 2 Report

The authors have addressed properly to my previous comments. I have no further questions.